# ALPHACON: IN-CONTEXT ADAPTATION FOR DYNAMIC ALPHA GENERATION

## ABSTRACT

Finding predictive signals known as alphas for stock returns is a central challenge in quantitative finance. This challenge is complicated by the non-stationary nature of financial markets. Conventional automated methods learn a single static model from historical data, and may perform poorly when market regimes shift. In this work, we reformulate this task as a problem of in-context adaptation. Our goal is to train a single universal model that can adapt its generation process to different market conditions at inference time. We introduce AlphaCon, a novel framework that uses recent data as context to guide alpha generation without requiring retraining. The model learns this adaptive capability through a specialized two-level training procedure, where an outer loop optimizes the context encoder across diverse historical market tasks, and an inner loop refines the generation agents within each task. The generation process itself is structured as a two-stage proposal and refinement loop enhanced by a learnable advice mechanism. We train the entire framework using reinforcement learning. Experiments show that AlphaCon trained once significantly outperforms strong baselines that require periodic retraining. This demonstrates robust performance across diverse market regimes.

## 1 INTRODUCTION

Quantitative finance involves using mathematical models to make investment decisions in financial markets. These markets include stocks traded on exchanges, where prices fluctuate daily based on supply, demand, and economic factors. A key task in this field is discovering predictive signals known as alphas Qian et al. (2007). An alpha is a mathematical formula that analyzes market data, such as stock prices and trading volumes, to forecast future returns, which are the percentage changes in stock prices over time. For example, a simple alpha might compute the average closing price over the past five days to identify short-term trends.

Researchers create various automated methods to find these alphas Shi et al. (2025). Genetic programming evolves formulas by simulating natural selection to identify effective ones Cui et al. (2021). More recently, methods based on reinforcement learning (RL) emerge Yu et al. (2023). In a typical RL setup, a generative model, such as a Transformer-based architecture, acts as the agent. Its action is to sequentially generate tokens (operators and market features) that form a candidate alpha expression. This generated alpha undergoes evaluation on historical data, where its predictive performance, often measured by the information coefficient, serves as a reward signal. The agent then refines its policy through trial and error, learning to construct more effective expressions over time.

However, a critical limitation unites these existing methods. They produce a single, static model. These models train on vast historical data and learn a fixed set of parameters designed to perform well on average across the past. This static nature becomes a significant drawback in financial markets, which are inherently non-stationary and constantly shift between regimes like bull and bear markets. A static model cannot incorporate new market information at inference time, leading to a well-known problem called alpha decay, where predictive signals lose their effectiveness over time Pénasse (2022). While a common workaround is to periodically retrain the model on recent data, this approach is computationally expensive and fundamentally flawed. The small amount of new data often leads to minimal updates, meaning the model continues to reflect historical averages rather than adapting to current market conditions.

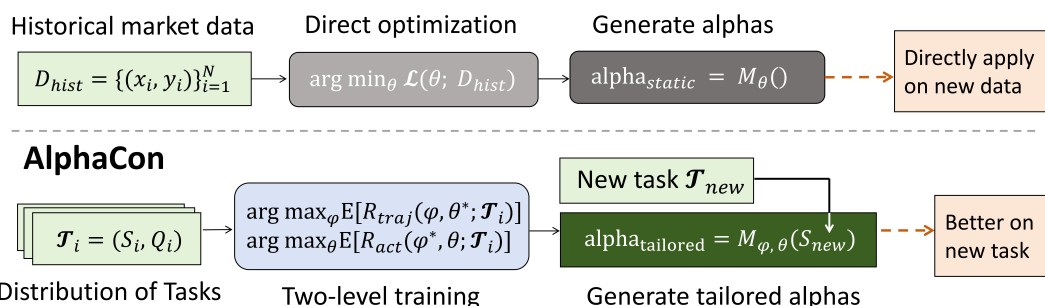

Figure 1: Conceptual comparison of the traditional approach versus our AlphaCon model. **(a) Traditional Approach:** A static model trains on all historical data and then applies directly to new data. This approach may perform poorly in new market regimes. **(b) AlphaCon:** Our model trains to learn an adaptive generation process. At inference time, it uses context from a new task ($S_{new}$) to generate a tailored alpha, leading to improved performance.

To address this challenge, we introduce AlphaCon, a novel framework that operationalizes in-context adaptation for alpha generation. We reformulate this challenge as a problem of in-context adaptation and build a universal model that, at inference time, takes recent market data as contextual input and generates alphas specifically tailored to that context. Instead of learning a fixed solution, AlphaCon learns an adaptive procedure, enabling it to rapidly specialize its behavior for new market environments without any retraining. As conceptually illustrated in Figure 1, AlphaCon's design fundamentally differs from traditional static approaches. At inference time, it first consumes recent market data from a new task to form a context, which then guides the generation of tailored alphas.

Furthermore, recognizing that effective alpha generation is a complex, reflective process, we decompose it into two stages of proposal and refinement. A Proposer LLM generates an initial draft alpha, which is then improved by a Refiner LLM. Crucially, this refinement is not a simple re-prompting. It receives guidance from a learnable advice mechanism that provides targeted, data-driven suggestions, making the process both structured and effective. The entire framework trains end-to-end with a specialized two-level reinforcement learning procedure that teaches the model how to adapt.

Our key empirical results demonstrate the superiority of this adaptive approach. We conduct extensive experiments across four major stock indices (CSI 300, CSI 500, NASDAQ 100, and S&P 500). A single, universally trained AlphaCon model, which adapts at inference time without any retraining, consistently outperforms strong baselines that require periodic retraining. On the CSI 300 and CSI 500 datasets, AlphaCon achieves an average improvement of over 10% in Information Coefficient (IC) and in Information Ratio (ICIR) over the next best baseline. This validates the effectiveness of learning an adaptive procedure over a static one.

Our main contributions are:

- We reformulate alpha discovery as an in-context adaptation problem, enabling a single model to adapt its behavior to unseen market regimes at inference time without retraining.

- We propose AlphaCon, a new model inspired by reflective LLM paradigms that employs a multi-stage process with separate agents for proposing and refining alphas.

- We design a two-level learning procedure that effectively trains the model to learn this adaptive capability.

- Extensive experiments show that a single, universally trained AlphaCon model significantly outperforms strong baselines that require periodic retraining, demonstrating robust and superior performance.

## 2 RELATED WORK

**Automated Alpha Generation.** The automated search for alphas initially relied heavily on Genetic Programming, evolving mathematical expressions often optimized using historical Information Coefficient (IC) Cui et al. (2021); Lin et al. (2019). Symbolic Regression, particularly Deep Symbolic Regression Petersen et al. (2021); Mundhenk et al. (2021), represents another automated avenue. More recent approaches have incorporated Reinforcement Learning. For instance, AlphaGen Yu et al. (2023) used RL to optimize a set of synergistic alphas, while AlphaForge Shi et al. (2025) introduced dynamic factor weighting. Despite their advancements, a common thread unites these approaches: they are designed to learn a single, *static model* from aggregate historical data. This fundamental design makes them inherently slow to adapt to the non-stationary dynamics of financial markets, as they lack an explicit mechanism to condition their behavior on the current market regime.

**Model Adaptation for Non-Stationary Environments.** A key challenge in dynamic settings is adapting models to concept drift. Many strategies address this reactively. Paradigms like transfer learning Jain et al. (2023); Kurmanji et al. (2024) and active learning Ma et al. (2020); Li et al. (2022) typically rely on periodic model retraining or fine-tuning to cope with data shifts. This process is often computationally slow and introduces significant latency. Our work takes a different path, aligning with context-based meta-learning. In this paradigm, the model learns an adaptive procedure that infers a task representation from recent data and uses this context to modulate its behavior at inference time, thereby avoiding gradient-based updates Rakelly et al. (2019); Zintgraf et al. (2019); Laskin et al. (2022).

## 3 PRELIMINARIES

We consider a market of $N$ stocks over a period of $T$ trading days. For each day $t$, we have access to market features $\mathbf{X}_t$ and the future returns $\mathbf{Y}_t$ which we aim to predict. An **alpha** is a function $f$ that maps historical market data to a vector of predictive scores, $\mathbf{v}_t = f(\mathbf{X}_t) \in \mathbb{R}^N$.

The quality of a single alpha is primarily measured by the **Information Coefficient (IC)**, which is the daily cross-sectional correlation between its scores and the actual returns:

$$\text{IC}_t(f) = \text{Correlation}(f(\mathbf{X}_t), \mathbf{Y}_t). \tag{1}$$

In practice, multiple alphas $\mathcal{F} = \{f_1, \ldots, f_k\}$ are combined to form a more stable portfolio signal. The performance of this set is evaluated by its **Portfolio IC**. This involves creating a composite signal by linearly combining the individual alpha values. The weights for this linear combination are determined by fitting a linear regression model on a given period of historical data. The resulting composite signal's correlation with returns is then calculated on a subsequent, unseen period.

Given the non-stationary nature of financial markets, we formulate the problem as learning an adaptive policy $\pi$. This policy should be capable of generating a specialized alpha portfolio $\mathcal{F}_i$ when conditioned on a specific market environment. We represent these market environments as a distribution of tasks $p(\mathcal{T})$, where each task $\mathcal{T}_i$ is a slice of historical data. The learning objective is to find the optimal policy $\pi^*$ that maximizes the expected performance over this task distribution. Formally, the policy takes the data from a task's support set $\mathcal{S}_i$ as input, and its output portfolio, $\pi(\mathcal{S}_i)$, is evaluated on the task's subsequent, unseen query set $\mathcal{Q}_i$. The overall objective is:

$$\pi^* = \arg\max_\pi \mathbb{E}_{(\mathcal{S}_i, \mathcal{Q}_i) \sim p(\mathcal{T})} \left[ \text{IC}_{\text{port}}(\pi(\mathcal{S}_i), \mathcal{Q}_i) \right]. \tag{2}$$

## 4 THE ALPHACON FRAMEWORK

### 4.1 OVERALL ARCHITECTURE

Learning to generate an entire portfolio of alphas in a single step is intractable. We therefore decompose this task into a sequential process, where one alpha is constructed at each step. This process is naturally modeled as a Markov Decision Process (MDP), enabling the use of reinforcement learning. Within each adaptive context, the agent's goal is to learn a policy that maximizes the cumulative reward, which is based on the marginal performance contribution of each new alpha.

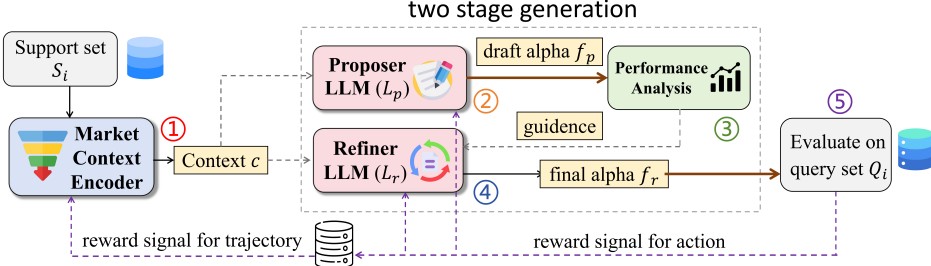

Figure 2: Overview of the AlphaCon framework. The model is composed of a Market Context Encoder and a two-stage alpha generation process. The modules are trained via a dual-loop RL method.

The AlphaCon architecture is designed to implement this adaptive process through a multi-stage workflow, as illustrated in Figure 2. The training of AlphaCon is organized into a two-level procedure. The first level handles task-level adaptation, which involves optimizing the modules that interpret the overall market context and provide strategic guidance. These include the Market Context Encoder and the performance analysis modules. The second level focuses on action-level generation, which updates the agents responsible for the concrete steps of creating alphas. These are the Proposer and Refiner LLMs. This dual optimization structure allows the model to learn both how to adapt to a new market and how to generate effective alphas within that context.

The alpha generation process for a specific market context unfolds in the following steps:

(1) **Market Context Encoding.** The process begins with the Market Context Encoder. This module analyzes historical market data from the input context to produce a compact conditioning vector **c**. This vector captures the current market regime and serves as the primary signal for adaptation.

(2) **Draft Proposal.** Conditioned on the context **c**, the first Generative Agent, the **Proposer LLM** ($L_p$), generates an initial draft alpha, $f_p$. This draft serves as a promising starting point for refinement.

(3) **Performance Analysis and Guidance.** AlphaConthen analyzes the draft. A diagnostic module evaluates the historical performance of $f_p$ on the contextual data. Based on this analysis, it retrieves targeted, learnable advice to guide the subsequent refinement step.

(4) **Guided Refinement.** The second Generative Agent, the **Refiner LLM** ($L_r$), takes the market context **c**, the draft alpha $f_p$, and the retrieved guidance to produce the final, improved alpha, $f_r$.

(5) **Portfolio Iteration.** The final alpha $f_r$ is added to the current portfolio. The process then repeats from Step 2 to generate the next alpha, progressively building a diverse and synergistic portfolio.

## 4.2 CONTEXT-CONDITIONED ALPHA PROPOSAL

The goal of the Market Context Encoder is to distill the high-dimensional, time-varying market data from a support set into a single, compact context vector **c**. A critical design choice here is to learn this representation end-to-end, rather than relying on pre-defined, low-dimensional market indicators (e.g., index volatility). Pre-defined indicators can be brittle and may miss subtle cross-sectional patterns or complex temporal dynamics. Our architecture is designed to capture these nuances. The parameters of this module are denoted by $\phi_E$.

The encoding process begins by transforming the daily market snapshot. For each day $t$ in the support set, we flatten the market data matrix $\mathbf{X}_t \in \mathbb{R}^{N \times F}$ into a single vector. A multi-layer perceptron (MLP) then projects this high-dimensional daily vector into a lower-dimensional embedding $\mathbf{x}'_t$:

$$\mathbf{x}'_t = \text{MLP}(\text{Flatten}(\mathbf{X}_t)) \in \mathbb{R}^d. \tag{3}$$

To capture the temporal dynamics, we use a Transformer encoder. This choice is motivated by its self-attention mechanism, which is uniquely suited to identify the most salient time steps within the support period that define the current market regime. It can learn to weigh, for instance, a

recent period of high volatility more heavily than a distant, quiet period, creating a more robust and informative context vector.

We assemble daily embeddings into a sequence $\{\mathbf{x}'_1, \ldots, \mathbf{x}'_{T_s}\}$. We then prepend a special, learnable [CLS] token embedding, $\mathbf{e}_{cls} \in \mathbb{R}^d$, to this sequence. Following standard practice in sequence modeling, the final hidden state corresponding to this token will serve as the aggregate representation for the entire sequence. This combined sequence is then processed by a Transformer encoder:

$$[\mathbf{h}_{cls}, \mathbf{h}_1, \ldots, \mathbf{h}_{T_s}] = \text{TransformerEnc}([\mathbf{e}_{cls}, \mathbf{x}'_1, \ldots, \mathbf{x}'_{T_s}]). \tag{4}$$

The output hidden state $\mathbf{h}_{cls}$ corresponding to the [CLS] token is designated as the final market context vector $\mathbf{c}$. This vector encapsulates the essential market characteristics and serves as the primary conditioning signal for the alpha generation process.

**Proposer LLM.** The Proposer, $L_p$, parameterized by $\theta_p$, is responsible for generating an initial alpha candidate. Conditioned on the market context $\mathbf{c}$ and the set of previously generated alphas $\mathcal{F}_{k-1}$, it models the probability of a draft alpha formula $f_p$:

$$f_p \sim P_{\theta_p}(f|\mathbf{c}, \mathcal{F}_{k-1}). \tag{5}$$

In practice, the context vector $\mathbf{c}$ is transformed into a soft prompt and prepended to the input of the LLM.

### 4.3 Performance Analysis and Guided Refinement

After the Proposer generates a draft alpha $f_p$, the framework performs a detailed analysis of its historical behavior. The goal is to provide the Refiner with not just the draft itself, but with actionable guidance on *how* to improve it. Simply providing scalar performance metrics (e.g., mean IC) is insufficient, as it hides crucial temporal dynamics. For instance, an alpha with a mean IC of 0.03 could be consistently stable or wildly volatile; each case requires a different refinement strategy. Our design explicitly addresses this by first diagnosing temporal patterns and then retrieving targeted advice.

The Diagnostic Encoder's goal is to create a structured representation of the alpha's characteristics. To do this, we compute two distinct diagnostic time series. The first is the daily Information Coefficient (IC) sequence, $\mathbf{s}_{ic} \in \mathbb{R}^{T_s}$, which measures the alpha's raw predictive power over time. The second is the daily correlation sequence, $\mathbf{s}_{corr} \in \mathbb{R}^{T_s}$, which measures the draft's redundancy with the existing alpha portfolio $\mathcal{F}_{k-1}$. We use a 1D-CNN to process these series because of its proven ability to learn and extract meaningful temporal motifs, such as performance decay, seasonality, or volatility spikes. This allows the model to move beyond simple averages and understand the behavioral signature of the draft alpha. A global pooling layer then aggregates these features into a final query vector $\mathbf{q}$:

$$\mathbf{q} = \text{GlobalPool}(\text{Conv1D}_{\phi_{DE}}([\mathbf{s}_{ic}; \mathbf{s}_{corr}])) \in \mathbb{R}^{d_{cnn}}. \tag{6}$$

This vector $\mathbf{q}$ serves as a structured query summarizing the key strengths and weaknesses of the draft alpha.

This query vector is then used to retrieve targeted feedback from the Refinement Advice Retriever, a learnable knowledge base of refinement strategies. The retriever is composed of two learnable components: a matrix of **Opinion Keys** ($\mathbf{M}_k \in \mathbb{R}^{N_c \times d_{cnn}}$) and a corresponding tensor of **Opinion Values** ($\mathbf{M}_v \in \mathbb{R}^{N_c \times L_v \times d}$). Each key vector can be thought of as learning to represent a canonical performance problem (e.g., high but decaying IC), while its corresponding value sequence encodes a piece of advice as a soft prompt (e.g., an abstract suggestion to "apply smoothing"). The retrieval is performed via an attention mechanism where the query $\mathbf{q}$ is compared against all keys to compute attention weights, $\{\alpha_i\}$.

$$\alpha_i = \text{softmax}_i \left( \frac{\mathbf{q} \cdot \mathbf{m}_{k,i}}{\sqrt{d_{cnn}}} \right). \tag{7}$$

These weights are used to compute a weighted sum of all opinion values, constructing the final advice vector $\mathbf{v} \in \mathbb{R}^{L_v \times d}$. This vector is a dynamic, context-specific soft prompt that provides tailored guidance to the Refiner LLM.

$$\mathbf{v} = \sum_{i=1}^{N_c} \alpha_i \mathbf{m}_{v,i}. \tag{8}$$

**Refiner LLM.** The Refiner, $L_r$, with parameters $\theta_r$, produces the final alpha $f_r$. It is conditioned on a richer information set: the market context $\mathbf{c}$, the draft alpha $f_p$, and the retrieved advice vector $\mathbf{v}$. Its policy is thus:

$$f_r \sim P_{\theta_r}(a|f_p, \mathbf{c}, \mathbf{v}, \mathcal{F}_{k-1}). \tag{9}$$

The vectors $\mathbf{c}$ and $\mathbf{v}$ are injected as soft prompts, enabling the refiner to perform a context-aware and advice-driven modification of the draft.

### 4.4 DUAL-LOOP TRAINING FRAMEWORK

The entire model is trained using a dual-loop procedure designed around a two-level optimization. The outer loop optimizes the modules responsible for task-level adaptation, enhancing their ability to provide effective context and guidance across different market tasks. Concurrently, the inner loop updates the agents responsible for action-level generation (the LLMs), improving their execution of the proposal and refinement steps within a given task.

**Reward Functions.** The generation of a portfolio is an MDP. To promote generalization across tasks, rewards are computed on an out-of-sample query set. Let $G(f, \mathcal{F}) = \text{IC}_{\text{port}}(\mathcal{F} \cup \{f\}) - \text{IC}_{\text{port}}(\mathcal{F})$ be the marginal gain in Portfolio IC. We define separate rewards for the two LLM agents. The Proposer reward, $R_p = G(f_r, \mathcal{F}_{k-1})$, is the marginal gain of the final, refined alpha. This delayed credit assignment encourages the Proposer to generate drafts with high potential for refinement, rather than settling for locally optimal but less improvable candidates. The Refiner reward, $R_r = G(f_r, \mathcal{F}_{k-1}) - G(f_p, \mathcal{F}_{k-1})$, measures the value added by the refinement process. This incremental formulation isolates the contribution of the Refiner. For the task-level adaptation, we define a trajectory reward $R_{\text{traj}}$ as the Portfolio IC of the final, complete alpha set $\mathcal{F}_K$.

**Inner Loop: Action-Level Optimization.** The inner loop updates the Proposer and Refiner LLMs at a high frequency. We use the PPO clipped surrogate objective to update the LoRA parameters of the LLMs. For a generic agent policy $\pi_\theta$, the objective is:

$$\mathcal{L}^{\text{CLIP}}(\theta) = \hat{\mathbb{E}}_t \left[ \min(r_t(\theta)\hat{A}_t, \text{clip}(r_t(\theta), 1 - \epsilon, 1 + \epsilon)\hat{A}_t) \right], \tag{10}$$

where $r_t(\theta)$ is the probability ratio and $\hat{A}_t$ is the estimated advantage. The Proposer's parameters $\theta_p$ are updated by maximizing Eq. 10 with rewards $R_p$, while the Refiner's parameters $\theta_r$ are updated using rewards $R_r$.

**Outer Loop: Task-Level Optimization.** The outer loop runs at a lower frequency to update the adaptation modules: the Market Encoder ($\phi_E$), Diagnostic Encoder ($\phi_{DE}$), and the Opinion Pool ($M_k, M_v$). We frame this as a task-level actor-critic problem Mnih et al. (2016), where the entire generative process serves as the actor. To stabilize training, a simple critic provides a baseline, which is implemented as a moving average of past rewards. The actor's parameters are updated using policy gradients, driven by an advantage signal calculated as the trajectory reward $R_{\text{traj}}$ minus this baseline. This approach allows the model to effectively learn what constitutes an informative market context and useful guidance for alpha generation.

## 5 EXPERIMENTS

This section presents a comprehensive empirical evaluation of the proposed AlphaCon framework. We aim to answer the following research questions:

- **RQ1:** How does the full AlphaCon framework perform against a wide range of strong baselines, including state-of-the-art alpha generation methods and a simplified "propose-only" variant?

- **RQ2:** What are the individual contributions of the core components, particularly the diagnostic module and the attention-based opinion pool that enable the refinement process?

- **RQ3:** Can a single, universally trained AlphaCon model, adapting purely at inference time, outperform strong models that are periodically retrained on new data, demonstrating true train-free adaptation?

## 5.1 EXPERIMENTAL SETTINGS

This section details the datasets, baselines, and evaluation metrics used. Further details are in Appendix B.

**Datasets.** We use data from four major stock indices, covering both Chinese A-shares (CSI 300, CSI 500) and U.S. markets (NASDAQ 100, S&P 500), to ensure a broad evaluation. For all markets, the data spans from January 1, 2012, to April 1, 2025, and is split chronologically into a training set (2012-2022), a validation set (2023), and a test set (Jan 2024 - Apr 2025). Raw features include daily Open, High, Low, Close, Volume, and VWAP. The prediction target is the next day's return.

**Baselines.** We compare AlphaCon against a range of automated alpha generation methods. These include: **GP**, implemented as in the AlphaGen baseline Yu et al. (2023); **DSO** Landajuela et al. (2022), a prominent symbolic regression approach; **AlphaGen** Yu et al. (2023), which utilizes reinforcement learning to find synergistic alphas; and **AlphaForge** Shi et al. (2025), a recent method based on a generative-predictive neural network. We also include a key ablation, **AlphaCon-PO** (propose-only), which removes the diagnose-and-refine loop to isolate the value of the reflection process.

**Evaluation Metrics.** We evaluate all models on the test set using standard metrics: **IC** (Information Coefficient) for raw predictive power; **RankIC**, its rank-based counterpart robust to outliers; **ICIR** (IC Information Ratio) for signal consistency; and its rank-based version, **RankICIR**. Higher values are better for all metrics.

## 5.2 OVERALL PERFORMANCE COMPARISON (RQ1)

To answer our first research question, we conduct a comprehensive comparison of the full AlphaCon framework against all baseline methods under a dynamic evaluation protocol that simulates real-world deployment. For this test, covering the period from Jan 2024 to Apr 2025, we update the models monthly. At the beginning of each month, AlphaCon uses the preceding quarter's data as context to adapt and generate a new alpha portfolio without any retraining. In contrast, all baseline models are fully retrained using all available historical data up to that point. The performance metrics reported in Table 1 are aggregated over the entire test period, reflecting the cumulative effectiveness of each model's adaptation or retraining strategy.

Table 1: Overall performance comparison on CSI 300, CSI 500, NASDAQ 100, and S&P 500 test sets (2024.1.1 - 2025.4.1). We report core signal quality and stability metrics. Bold values indicate the best performance for each metric, and underlined values indicate the second best.

| Model | CSI 300 | | | | CSI 500 | | | |
|---|---|---|---|---|---|---|---|---|
| | IC | RankIC | ICIR | RankICIR | IC | RankIC | ICIR | RankICIR |
| GP | 0.0245 | 0.0373 | 0.1153 | 0.1893 | 0.0192 | 0.0385 | 0.1200 | 0.2217 |
| DSO | 0.0301 | 0.0472 | 0.1250 | 0.1937 | 0.0211 | 0.0423 | 0.1328 | 0.2376 |
| AlphaGen | 0.0365 | 0.0572 | 0.1354 | 0.2157 | 0.0230 | 0.0468 | 0.1416 | 0.2453 |
| AlphaForge | 0.0371 | 0.0589 | 0.1453 | 0.2124 | 0.0227 | 0.0463 | 0.1523 | 0.2855 |
| AlphaCon-PO | 0.0382 | 0.0601 | 0.1440 | 0.2110 | 0.0251 | 0.0471 | 0.1538 | 0.2618 |
| **AlphaCon(Full)** | **0.0425** | **0.0631** | **0.1755** | **0.2312** | **0.0261** | **0.0518** | **0.1805** | **0.2860** |

| Model | NASDAQ 100 | | | | S&P 500 | | | |
|---|---|---|---|---|---|---|---|---|
| | IC | RankIC | ICIR | RankICIR | IC | RankIC | ICIR | RankICIR |
| GP | 0.0105 | 0.0115 | 0.0710 | 0.0850 | 0.0082 | 0.0135 | 0.0611 | 0.0913 |
| DSO | 0.0112 | 0.0125 | 0.0750 | 0.0890 | 0.0089 | 0.0141 | 0.0654 | 0.0952 |
| AlphaGen | 0.0135 | 0.0152 | 0.1083 | 0.1156 | 0.0115 | 0.0163 | **0.0985** | 0.1088 |
| AlphaForge | 0.0141 | 0.0148 | 0.1151 | 0.1251 | 0.0109 | 0.0171 | 0.0853 | 0.1121 |
| AlphaCon-PO | 0.0153 | 0.0164 | 0.1142 | 0.1240 | 0.0112 | 0.0168 | 0.0921 | 0.1105 |
| **AlphaCon(Full)** | **0.0173** | **0.0184** | **0.1262** | **0.1312** | **0.0128** | **0.0186** | 0.0951 | **0.1198** |

The results in Table 1 show that the AlphaCon(Full) framework achieves strong performance across the four diverse datasets. It obtains the highest values on most metrics for the CSI 300, CSI500, and NASDAQ 100 markets. On the S&P 500, its performance is also highly competitive, securing either the best or second-best results on all reported metrics. The performance gap between the full model and the AlphaCon-PO ablation highlights the contribution of the two-stage process. While baselines like AlphaGen and AlphaForge demonstrate solid performance, they are generally outperformed by AlphaCon, which points to the potential benefits of an adaptive, multi-stage generation approach.

## 5.3 ABLATION STUDY (RQ2)

To investigate the individual contributions of the key components within the AlphaCon framework, we conduct an extensive ablation study. We compare the full model against several variants, each designed to isolate and evaluate a specific architectural choice. The variants are:

- **AlphaCon-PO:** Removes the entire diagnose-and-refine loop, serving as the baseline for the value of reflection.
- **w/o Diagnostic Encoder:** Replaces the 1D-CNN encoder with a simple MLP that operates on scalar metrics (mean IC and mean correlation), removing the model's ability to process temporal patterns.
- **w/o Advice Vector:** Removes the Attention-based Opinion Pool entirely. The Refiner LLM must attempt to improve the draft alpha without any explicit external advice.
- **w/ Static Advice:** Replaces the dynamic, attention-based advice with a single, learnable static soft prompt, providing the same generic advice for every situation.
- **w/ Single LLM:** Uses a single LLM with shared LoRA weights to perform both the proposing and refining tasks, testing the benefit of agent specialization.
- **w/ Flat Optimization:** Replaces the two-level training procedure with a single-loop optimization, where all model components are updated at the same frequency using action-level rewards, testing the necessity of the structured training framework.

All variants are evaluated on the CSI300 test set, and the results are presented in Table 2.

Table 2: Ablation study results on the CSI300 test set. Performance metrics quantify the contribution of each removed or modified component.

| Variant | IC | RankIC | ICIR | RankICIR |
|---|---|---|---|---|
| **AlphaCon(Full)** | **0.0425** | **0.0631** | **0.1755** | **0.2312** |
| AlphaCon-PO | 0.0382 | 0.0601 | 0.1440 | 0.2110 |
| w/o Diagnostic Encoder | 0.0405 | 0.0608 | 0.1655 | 0.2108 |
| w/o Advice Vector | 0.0384 | 0.0592 | 0.1587 | 0.2173 |
| w/ Static Advice | 0.0392 | 0.0587 | 0.1610 | 0.2115 |
| w/ Single LLM | 0.0415 | 0.0617 | 0.1732 | 0.2246 |
| w/ Flat Optimization | 0.0388 | 0.0595 | 0.1503 | 0.2118 |

The results in Table 2 confirm that each component contributes positively to the final performance. The performance gap between the full model and AlphaCon-PO establishes the significant value of the entire reflection process. Moreover, removing the advice mechanism entirely (w/o Advice Vector) degrades performance to a level only slightly better than not refining at all. Providing generic, static advice (w/ Static Advice) offers a marginal benefit, but is significantly outperformed by the full model's ability to retrieve tailored advice. This suggests that the model benefits not just from advice in general, but specifically from advice that is dynamically conditioned on the problem.

Furthermore, the quality of the diagnostic input is critical. The 'w/o Diagnostic Encoder' variant, which cannot process temporal patterns in the alpha's performance, shows a noticeable drop in all metrics. This indicates that the diagnostic module's ability to capture time-series dynamics is key to forming an effective query for the advice retriever. Finally, the performance degradation of the 'w/ Flat Optimization' variant confirms that the task-level adaptation modules require a stable, trajectory-level reward signal for effective optimization, as the more frequent action-level rewards are too noisy.

## 5.4 Long-Term Adaptability(RQ3)

To evaluate the core claim of train-free adaptation, particularly its robustness during shifting market conditions, we design a long-term backtest experiment from January 1, 2021, to December 31, 2024. This four-year out-of-sample period contains multiple distinct market regimes, including significant bull and bear phases, serving as a robust test of a model's adaptability under non-stationarity. Similar to the experiment settings in section 5.2, the AlphaCon is trained only once on all available data prior to the backtest period (i.e., before Jan 1, 2021), and the baseline models follow the standard periodic retraining paradigm. We also include the **CSI 300 Index** as a passive market benchmark.

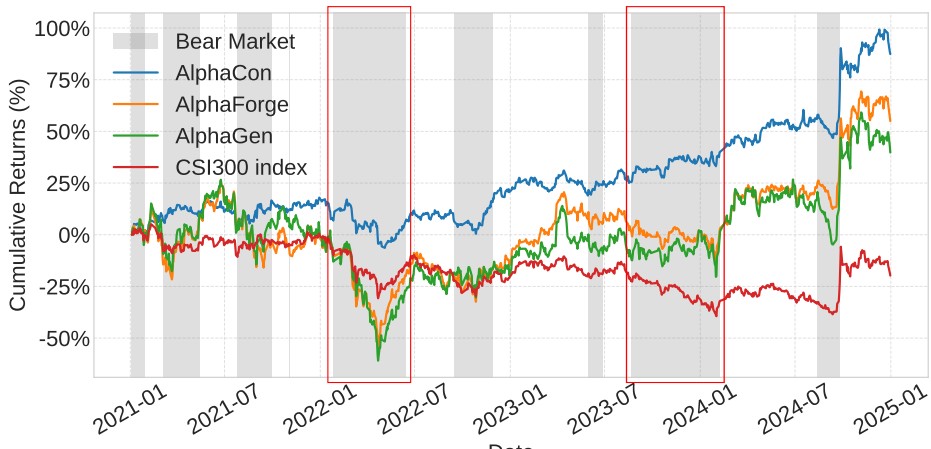

Figure 3: Cumulative portfolio returns over a four-year backtest period (2021-2024) on CSI300. AlphaCon(trained once, adapted monthly) is compared against baselines (retrained monthly) and the market index. Shaded regions are bear markets

To analyze performance during market shifts, we simulate a daily long-only portfolio strategy for each model. The cumulative returns, shown in Figure 3, are calculated by holding the top 10% of stocks ranked by the model's alpha scores each day. We identify bear market periods on the graph, marked by shaded regions, defined by periods where the CSI 300 index's closing price is below 95% of its 120-day simple moving average.

The results show that while all strategies beat the market, AlphaCon achieves the highest cumulative return with greater stability. A notable divergence occurs in bear markets (e.g., H1 2022 and H2 2023), where AlphaCon effectively avoids major drawdowns that plague the periodically retrained baselines. This performance gap suggests a fundamental difference in adaptation mechanisms. Periodically retrained models, which optimize over vast historical datasets, tend to produce generalized, average solutions that are slow to react to regime shifts. In contrast, AlphaCon's in-context adaptation explicitly conditions its generation process on recent market data, which allows it to produce a policy more attuned to the current environment.

## 6 Conclusion

In this paper, we present AlphaCon, a framework that reframes alpha discovery as an in-context adaptation problem. By decomposing alpha generation into a proposal and refinement process, AlphaCon learns an adaptive procedure rather than a static solution. Our experiments demonstrate that a single, universally trained model can significantly outperform periodically retrained baselines, maintaining robust performance across diverse and shifting market regimes. This work opens up new avenues for building truly dynamic and intelligent systems in quantitative finance. Future research can extend this paradigm by exploring more sophisticated adaptive mechanisms and dynamically expanding the creative space of the generative agents.

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

## A PREDEFINED OPERATORS FOR ALPHA GENERATION

In the AlphaCon framework, the LLM generates alpha factors as mathematical formulas. These formulas are constructed using a predefined set of operators, largely based on the functionalities available in the Qlib library. This appendix lists the primary operators available to the LLM. Operands for these functions can be raw market features (e.g., denoted as '$close', '$open', '$high', '$low', '$volume', '$vwap'), numerical constants, or the results of other operator expressions.

### A.1 BASIC DATA ACCESS

- `$feature_name`: Accesses the time-series data for a given raw feature (e.g., `$close` for closing prices, `$volume` for trading volume).

### A.2 OPERATOR TABLE

The following table details the common operators used for formulaic alpha generation. In the table, 'x', 'y', 'z' typically represent feature expressions or constants. 'N' usually denotes a lookback window (integer), and 'q' represents a quantile value (0-1). For rolling operators, if 'N=0', it often implies an expanding window from the beginning of the series.

Table 3: Predefined Qlib-based Operators

| Operator (Syntax) | Description | Parameters |
|---|---|---|
| **Element-Wise (Unary) Operators** | | |
| Abs(x) | Absolute value of 'x'. | 'x' |
| Sign(x) | Sign of 'x' (-1, 0, or 1). | 'x' |
| Log(x) | Natural logarithm of 'x'. | 'x' |
| Not(x) | Logical NOT of 'x' (for boolean series). | 'x' |
| **Pair-Wise (Binary) Operators** | | |
| Add(x, y) | Element-wise sum: 'x + y'. | 'x', 'y' |
| Sub(x, y) | Element-wise difference: 'x - y'. | 'x', 'y' |
| Mul(x, y) | Element-wise product: 'x * y'. | 'x', 'y' |
| Div(x, y) | Element-wise division: 'x / y'. | 'x', 'y' |
| Power(x, y) | Element-wise power: 'x' raised to 'y'. | 'x' (base), 'y' (exponent) |
| Greater(x, y) | Element-wise maximum of 'x' and 'y'. | 'x', 'y' |
| Less(x, y) | Element-wise minimum of 'x' and 'y'. | 'x', 'y' |
| Gt(x, y) | 'x > y' (boolean). | 'x', 'y' |
| Ge(x, y) | 'x >= y' (boolean). | 'x', 'y' |
| Lt(x, y) | 'x < y' (boolean). | 'x', 'y' |
| Le(x, y) | 'x <= y' (boolean). | 'x', 'y' |
| Eq(x, y) | 'x == y' (boolean). | 'x', 'y' |
| Ne(x, y) | 'x != y' (boolean). | 'x', 'y' |
| And(x, y) | Logical AND of 'x' and 'y' (boolean). | 'x', 'y' |
| Or(x, y) | Logical OR of 'x' and 'y' (boolean). | 'x', 'y' |
| **Conditional Operator (Ternary)** | | |
| If(cond, x, y) | If 'cond' is true, then 'x', else 'y'. | 'cond' (boolean), 'x', 'y' |
| **Time-Series / Rolling Operators (Unary with Window)** | | |
| | | Continued on next page |

Table 3: Predefined Qlib-based Operators (Continued)

| Operator (Syntax) | Description | Parameters |
|---|---|---|
| Ref(x, N) | Value of 'x' from 'N' periods ago. (Lag) | 'x', 'N' |
| Mean(x, N) | Rolling mean of 'x' over 'N' periods. | 'x', 'N' |
| Sum(x, N) | Rolling sum of 'x' over 'N' periods. | 'x', 'N' |
| Std(x, N) | Rolling standard deviation of 'x' over 'N' periods. | 'x', 'N' |
| Var(x, N) | Rolling variance of 'x' over 'N' periods. | 'x', 'N' |
| Skew(x, N) | Rolling skewness of 'x' over 'N' periods (N>=3). | 'x', 'N' |
| Kurt(x, N) | Rolling kurtosis of 'x' over 'N' periods (N>=4). | 'x', 'N' |
| Max(x, N) | Rolling maximum of 'x' over 'N' periods. | 'x', 'N' |
| IdxMax(x, N) | Index of rolling maximum of 'x' over 'N' periods. | 'x', 'N' |
| Min(x, N) | Rolling minimum of 'x' over 'N' periods. | 'x', 'N' |
| IdxMin(x, N) | Index of rolling minimum of 'x' over 'N' periods. | 'x', 'N' |
| Quantile(x, N, q) | Rolling 'q'-th quantile of 'x' over 'N' periods. | 'x', 'N', 'q' |
| Med(x, N) | Rolling median of 'x' over 'N' periods. | 'x', 'N' |
| Mad(x, N) | Rolling Mean Absolute Deviation of 'x' over 'N' periods. | 'x', 'N' |
| Rank(x) | Cross-sectional rank (percentile) of 'x'. | 'x' |
| Rank(x, N) | Rolling rank (percentile) of 'x' over 'N' periods. | 'x', 'N' |
| Count(x, N) | Rolling count of non-NaN values in 'x' over 'N' periods. | 'x', 'N' |
| Delta(x, N) | Difference: 'x_t - x_t-N'. | 'x', 'N' |
| Slope(x, N) | Rolling linear regression slope of 'x' (vs. time index) over 'N' periods. | 'x', 'N' |
| Rsquare(x, N) | Rolling R-squared of 'x' (vs. time index) over 'N' periods. | 'x', 'N' |
| Resi(x, N) | Rolling residuals of 'x' (vs. time index) over 'N' periods. | 'x', 'N' |
| WMA(x, N) | Weighted Moving Average of 'x' over 'N' periods. | 'x', 'N' |
| EMA(x, N) | Exponential Moving Average of 'x'. If '0<N<1', 'N' is alpha; otherwise 'N' is span. | 'x', 'N' |
| **Pair-Wise Rolling Operators (Binary with Window)** | | |
| Corr(x, y, N) | Rolling correlation of 'x' and 'y' over 'N' periods. | 'x', 'y', 'N' |
| Cov(x, y, N) | Rolling covariance of 'x' and 'y' over 'N' periods. | 'x', 'y', 'N' |
| **Specialized Operators** | | |
| TResample(x, freq, method) | Resamples 'x' to frequency 'freq' using 'method'. | 'x', 'freq' (e.g., 'M', 'W'), 'method' (e.g., 'mean', 'sum', 'last') |

Note that the 'Rank' operator can be used in two ways:

- Rank(x): Computes the cross-sectional rank (typically as a percentile from 0 to 1) of feature 'x' across all stocks at each point in time.

- `Rank(x, N)`: Computes the rolling time-series rank (percentile) of feature 'x' for each stock over the preceding 'N' periods.

The precise behavior of these operators, especially concerning 'NaN' handling and 'min_periods' for rolling functions, generally follows the conventions of the Pandas library, which Qlib builds upon. The LLM is guided to combine these operators and operands to form candidate alpha factors. For example, `Rank(Correlation(Rank($volume), Rank($close), 5))` represents a momentum-like signal.

## B  IMPLEMENTATION DETAILS

This appendix provides supplementary details regarding the experimental setup, model architectures, and training procedures used in this paper to ensure reproducibility.

### B.1  TASK CONSTRUCTION

To create the distribution of tasks $p(\mathcal{T})$, we employed a sliding window approach over the training dataset (Jan 1, 2012 - Dec 31, 2022). Each task $\mathcal{T}_i = (\mathcal{S}_i, \mathcal{Q}_i)$ is constructed as follows:

- **Support Set ($\mathcal{S}_i$):** A period of trading days in a quarter (approx. 63 days) is used as the support set. The Market Context Encoder and the Diagnostic Module operate on data from this period.

- **Query Set ($\mathcal{Q}_i$):** The subsequent month (approx. 20 days) immediately following the support set is designated as the query set. All rewards for the RL agents are calculated based on performance on this out-of-sample data.

- **Sampling:** We slid this 4-month window across the entire training dataset with a stride of 1 month.

### B.2  MODEL ARCHITECTURE DETAILS

- **Market Context Encoder:** The input daily market data $\mathbf{X}_t \in \mathbb{R}^{N \times F}$ (where $N$ is the number of stocks and $F = 6$ is the number of features) is first flattened. The subsequent MLP consisted of two linear layers with a ReLU activation, projecting the flattened vector to a dimension of $d = 512$. The Transformer encoder is composed of 4 layers, each with 8 attention heads and a feed-forward network dimension of 2048. The final context vector $\mathbf{c}$ has a dimension of $d = 512$.

- **LLM Agents (Proposer & Refiner):** Both agents are based on the pre-trained `Llama-3-8B-Instruct` model. To make training computationally feasible, we employed the Low-Rank Adaptation (LoRA) technique. We applied LoRA with a rank of $r = 16$ and an alpha of $\alpha = 32$ to the query ('q_proj') and value ('v_proj') matrices of the attention mechanism. The market context vector $\mathbf{c}$ is transformed into a soft prompt of length 8, and the advice vector $\mathbf{v}$ is a soft prompt of length 4.

- **Diagnostic Encoder:** The input to the 1D-CNN is a two-channel time series of length $T_s = 63$ (daily ICs and correlations). The network consisted of three convolutional layers with kernel size 5 and stride 2. The number of channels increased from 2 to 16, then 32, and finally 64. A global average pooling layer is applied to the output of the final convolutional layer, producing a query vector $\mathbf{q}$ of dimension $d_{cnn} = 64$.

- **Refinement Advice Retriever:** The retriever's knowledge base contained $N_c = 16$ opinion key-value pairs. The Opinion Values tensor $\mathbf{M}_v$ had a shape of $16 \times 4 \times 512$ (to produce a soft prompt of length 4 with embedding dimension 512).

## C  LLM AGENT PROMPTS

This section details the prompt architecture for the Proposer and Refiner LLMs. A key aspect of our framework is the use of **soft prompts** for conditioning. These are learnable vector sequences, not human-readable text, derived from context vectors. In the templates below, we conceptually mark their injection points. The LLM's task is defined by the textual user prompt, which constrains its behavior and output format.

## C.1 PROPOSER LLM PROMPT

The Proposer is conditioned on the market context via a soft prompt. Its task is to generate a promising draft alpha that is novel compared to the existing portfolio.

---

**System Prompt**

You are an expert quantitative analyst. Your goal is to create novel, syntactically correct alpha factor formulas. Your output must be a single JSON object.

---

**User Prompt**

'[SOFT PROMPT FOR MARKET CONTEXT IS PREPENDED HERE]'

**Existing Alpha Portfolio:** [ existing_alphas_list ]

**Allowed Operators and Features:** - **Operators:** 'Mean(x, N)', 'Std(x, N)', 'Rank(x)', 'Corr(x, y, N)', 'Delta(x, N)', 'Abs(x)', 'Log(x)', 'Add(x, y)', 'Sub(x, y)', 'Mul(x, y)', 'Div(x, y)', 'If(cond, x, y)', etc. (A complete list is in Appendix A). - **Features:** '$open$', 'high', '$low$', 'close', '$volume$', 'vwap'.

**Task:** 1. Based on the provided market context, propose a NEW alpha formula. 2. The formula must be innovative and complementary to the existing portfolio. 3. The formula must strictly adhere to the allowed syntax, operators, and features. 4. Provide a brief rationale for your proposal.

**Output Format (JSON only):** "'json "draft_alpha": "<Your proposed alpha formula string>", "rationale": "<A brief explanation of why this alpha might work>" "'

---

## C.2 REFINER LLM PROMPT

The Refiner is conditioned on two soft prompts: one for the market context and one containing the refinement advice. Its task is to modify and improve a given draft alpha.

---

**System Prompt**

You are an expert quantitative researcher. Your goal is to analyze and refine existing alpha factors based on performance diagnostics. Your output must be a single JSON object.

---

**User Prompt**

'[SOFT PROMPT FOR MARKET CONTEXT IS PREPENDED HERE]' '[SOFT PROMPT FOR REFINEMENT ADVICE IS PREPENDED HERE]'

**Existing Alpha Portfolio:** [ existing_alphas_list ]

**Draft Alpha to Refine:** "draft_alpha"

**Allowed Operators and Features:** - **Operators:** 'Mean(x, N)', 'Std(x, N)', 'Rank(x)', 'Corr(x, y, N)', 'Delta(x, N)', 'Abs(x)', 'Log(x)', 'Add(x, y)', 'Sub(x, y)', 'Mul(x, y)', 'Div(x, y)', 'If(cond, x, y)', etc. (A complete list is in Appendix A). - **Features:** '$open$', 'high', '$low$', 'close', '$volume$', 'vwap'.

**Task:** 1. Analyze the provided draft alpha in light of the diagnostic advice (delivered via soft prompt). 2. Modify the formula to address the performance issues or enhance its strengths. 3. The refined formula must strictly adhere to the allowed syntax. 4. Provide a clear rationale explaining how your refinement addresses the performance diagnostics.

---

**Output Format (JSON only):** "'json  "refined_alpha": "<Your refined alpha formula string>", "refinement_rationale": "<A clear explanation of your changes>"  "'

In the above templates, existing_alphas_list and draft_alpha are placeholders for the respective alpha formula strings, which are injected as plain text.

