# OpenReview forum: "AlphaCon: In-Context Adaptation for Dynamic Alpha Generation"
_ICLR.cc/2026/Conference — ICLR 2026 Conference Withdrawn Submission_

### Official Review · Reviewer_NJP1 · 2025-10-28

**Soundness:** 2
**Presentation:** 2
**Contribution:** 2
**Rating:** 4
**Confidence:** 3

**Summary:**

This paper proposes AlphaCon, an in-context adaptation framework for dynamic alpha generation in quantitative finance. Instead of training static models that quickly lose predictive power as market regimes shift, AlphaCon learns a universal policy that adapts to new market conditions at inference time without retraining. The model integrates a Market Context Encoder and a two-stage generation process consisting of a Proposer LLM and a Refiner LLM, where refinement is guided by a learnable advice mechanism derived from diagnostic performance analysis. Trained through a dual-loop reinforcement learning structure that separates task-level adaptation and action-level optimization, AlphaCon achieves significantly better IC and ICIR scores across CSI300, CSI500, NASDAQ100, and S&P500 benchmarks.

**Strengths:**

1. The motivation of the paper is clear. It reformulates the problem of alpha factor discovery in quantitative investment as an in-context adaptation task, representing the first attempt to dynamically generate factors based on recent market data without retraining.
2. Drawing inspiration from reflective or critic-based paradigms in large language models, the paper introduces a two-stage generation framework composed of a Proposer and a Refiner LLM, together with a learnable advice mechanism that enables a reflection-style refinement process.
3. The experiments cover both Chinese and U.S. markets, demonstrating the model’s stable adaptability across different regions and market regimes, while the selected baselines are relatively up-to-date and competitive.

**Weaknesses:**

1. The experimental section mainly presents performance metrics but lacks interpretability analysis of model behavior, such as how the context encoder captures market states or whether the Refiner’s modifications align with financial intuition. There should be some intuitive statistical analyses or at least a case study to illustrate these aspects.
2. Although multiple baselines are included, the paper does not compare against recent large-model-based alpha generation methods or conventional machine learning approaches, which weakens the persuasiveness of the results.
3. The *Preliminaries* section is insufficiently detailed, making it difficult for readers to quickly grasp the core task being addressed.
4. The methodological description is not very clear; while the inputs and outputs of each module are roughly defined, they are not explicitly specified. Adding a case study that clarifies the data flow at each step could significantly improve readability.
5. Although the appendix provides some example prompts, the paper does not release implementation code, which may affect the reproducibility and credibility of the work.
6. The paper lacks ablation experiments on different LLM backbones, such as replacing Llama-3-8B-Instruct with other models to assess robustness.
7. There are several typographical errors, for example “AlphaConthen” in line 192.

**Questions:**

1. Is the market state captured by the context encoder interpretable? Could the authors provide evidence showing how context vectors differ across various market regimes?
2. Is the learnable advice mechanism shared across different markets, and if so, would large discrepancies in market styles (e.g., between Chinese and U.S. markets) lead to degraded performance? Additionally, is the advice itself interpretable?
3. The paper only reports *Cumulative portfolio returns over a four-year backtest period (2021–2024) on CSI300*. Are there corresponding results for the S&P 500 dataset?
4. Does the dual-loop reinforcement learning framework of AlphaCon have any theoretical guarantees or empirical analyses regarding its convergence and stability?

---

### Official Review · Reviewer_Z6hM · 2025-10-31

**Soundness:** 2
**Presentation:** 3
**Contribution:** 3
**Rating:** 4
**Confidence:** 3

**Summary:**

This paper proposes AlphaCon, a framework that reformulates automated alpha generation in quantitative finance as an in-context adaptation problem. Instead of retraining on new market data, AlphaCon uses recent market information as context to guide the generation of new alphas. The framework employs a dual-agent design—a Proposer and a Refiner Large Language Model—enhanced by a diagnostic encoder and a learnable advice mechanism. Training follows a two-level reinforcement learning procedure: an inner loop for agent-level updates and an outer loop for task-level adaptation. Experiments across four major stock indices (CSI 300, CSI 500, NASDAQ 100, and S&P 500) show that a single trained AlphaCon model outperforms baselines such as AlphaGen, AlphaForge, and symbolic regression methods, while maintaining robustness under non-stationary market conditions.

**Strengths:**

The paper presents a creative and conceptually appealing reformulation of the alpha discovery task, bridging ideas from meta-learning, reinforcement learning, and financial signal generation. The integration of context encoding, dual-agent reflection, and advice-based refinement is well-structured and resonates with the broader research direction of in-context adaptation. The design of the dual-loop optimization (outer task-level vs. inner action-level) demonstrates a solid grasp of meta-RL principles. The empirical evaluation is extensive, covering multiple markets and including ablation studies that isolate contributions of each module. The improvement over retrained baselines is notable, suggesting that AlphaCon indeed captures adaptable behavior.

**Weaknesses:**

While the motivation is clear, the conceptual novelty is overstated relative to existing meta-RL and in-context learning literature. The framework largely combines known ideas—context encoding, PPO-based RL, LoRA fine-tuning, and advice retrieval—without clear evidence of a new algorithmic principle. The claim of “train-free adaptation” relies entirely on conditioning mechanisms rather than demonstrated rapid generalization across unseen environments. Moreover, the experiments lack sufficient statistical rigor: no confidence intervals or significance tests are reported, and performance improvements (e.g., IC gains of 0.004–0.006) could be within noise levels in financial data.
The evaluation protocol also raises concerns: the model uses recent data as context, but it is unclear how lookahead bias is avoided when constructing the support/query sets. The description of task sampling (quarterly windows with monthly stride) may induce overlap and leakage.
Finally, the paper’s positioning within the financial AI literature feels underdeveloped—there is minimal discussion of how AlphaCon compares to domain-specific adaptive methods like online learning or Bayesian portfolio updates. The work would benefit from a clearer articulation of why in-context adaptation offers distinct advantages beyond reduced retraining frequency.

**Questions:**

How does AlphaCon handle outlier market conditions (e.g., sudden shocks) where the contextual quarter may not represent the upcoming period? Does the model degrade gracefully or overfit to short-term noise?

The use of LoRA-adapted LLMs is interesting but raises reproducibility concerns. Can the authors clarify the computational budget and whether smaller-scale models achieve similar performance?

Please provide statistical tests or bootstrapped confidence intervals to validate the observed improvements.

Clarify the temporal split between support and query sets to ensure no forward-looking bias.

Discuss computational efficiency—the paper argues that retraining is costly, but AlphaCon’s two-level RL with multiple agents seems expensive. Quantifying runtime trade-offs would make the claim more credible.

Consider comparing AlphaCon with more adaptive baselines (e.g., continual learning, model-agnostic meta-learning (MAML), or transformer-based financial forecasters).

The qualitative examples of generated alpha formulas or their interpretability would strengthen the paper and demonstrate the model’s creative capability.

---

### Official Review · Reviewer_YXb2 · 2025-11-02

**Soundness:** 2
**Presentation:** 2
**Contribution:** 2
**Rating:** 4
**Confidence:** 4

**Summary:**

The paper introduces AlphaCon, a framework for in-context adaptation in quantitative finance, specifically for dynamic alpha generation—the discovery of predictive trading signals. Instead of retraining static models as markets shift, AlphaCon learns a universal adaptive model that tailors alpha generation at inference time using recent market data as context.

**Strengths:**

I want to refrain from making strength comments before I can get more information on the computation side and code.

**Weaknesses:**

Limited theoretical analysis: While empirical results are strong, theoretical justification of convergence or generalization bounds for the dual-loop setup is absent.

Computational details: No runtime or resource comparison versus retraining baselines (important for practical deployment).

Reproducibility: Although appendices describe architecture, code release or pseudocode for the RL loops would improve transparency.

Interpretability: While the “advice vector” is conceptually interesting, no qualitative examples of retrieved advice are shown; visualizing this could strengthen the narrative.

**Questions:**

Could the authors report variance across multiple random seeds for the IC/ICIR metrics?

How does AlphaCon handle extreme market shocks (e.g., March 2020-type data)?

Would fine-tuning AlphaCon on a small amount of new data outperform pure in-context adaptation, or does that degrade generalization?

---

### Official Review · Reviewer_jjcV · 2025-11-04

**Soundness:** 2
**Presentation:** 2
**Contribution:** 1
**Rating:** 2
**Confidence:** 4

**Summary:**

This paper proposes AlphaCon, a novel in-context adaptation framework for automated alpha generation—the discovery of predictive signals for stock returns. Traditional reinforcement-learning or symbolic-regression-based alpha-mining methods train a static model from historical data and must be periodically retrained to cope with market regime shifts. AlphaCon instead aims to train a single universal model that dynamically adapts its behavior at inference time using recent market data as context.

The framework combines:

1. A Market Context Encoder that summarizes recent market features into a compact latent representation;

2. A two-stage alpha generation process involving a Proposer LLM (for initial drafts) and a Refiner LLM (for guided improvement);

3. A learnable advice mechanism that provides data-driven refinement prompts derived from a Diagnostic Encoder analyzing alpha performance; and

4. A dual-loop reinforcement-learning optimization, with an outer loop for task-level adaptation and an inner loop for action-level optimization.

Empirically, AlphaCon demonstrates significant gains over strong baselines (Genetic Programming, Deep Symbolic Regression, AlphaGen, AlphaForge) across four major stock indices (CSI 300/500, NASDAQ 100, S&P 500). Notably, it outperforms periodically retrained models by more than 10 % in IC and ICIR while requiring no retraining, and maintains superior robustness during non-stationary market regimes (2021–2024 backtest).

**Strengths:**

1. **Relevance to practice**: Addresses a central real-world, domain-specific issue -- _alpha decay_ -- by allowing inference-time adaptation without retraining.

2. Strong **Empirical Results**: The method consistently outperforms strong, modern baselines across four different major stock indices. The reported improvements of over 10% in IC and ICIR are substantial in the field of quantitative finance.

**Weaknesses:**

The chosen name of the framework: "Alpha**Con**" is telling. This paper reads more like an amalgamation of existing techniques drawn from a wide range of ML/AI disciplines (e.g., robotics [1, 2]) rather than a genuinely novel framework. Many of the ideas presented—particularly in-context adaptation and dual-level learning—have been extensively explored under different names and settings, and the paper appears to repackage these concepts with new terminology.

**Primary weaknesses**:

1. Inadequate **'Related Work**:
The discussion of related literature is insufficient. In-context adaptation using dual-stage or hierarchical processes at inference time has a vast and mature body of research. A more comprehensive review of such prior work would not only situate this paper properly within the literature but also clarify how its contributions differ or extend beyond established paradigms.

2. **Overstated Novelty and Contributions**: Several of the core contributions claimed by the authors are applications of well-known concepts, both in general and within the financial domain.

_On the claim_:
>We reformulate alpha discovery as an in-context adaptation problem, enabling a single model to
adapt its behavior to unseen market regimes at inference time without retraining.

While the paper frames this as a novel reformulation, (LLM) inference-time adaptation is a well-explored concept in machine learning, with prominent examples in fields like robotics [1, 2]. More importantly, this approach has also been specifically applied in the financial domain for adaptive alpha generation, as seen in recent work [3].


_On the claim_:
> We design a two-level learning procedure that effectively trains the model to learn this adaptive
capability

The claim that the paper “reformulates alpha discovery as an in-context adaptation problem” is overstated. Inference-time adaptation is neither novel in general nor unique to the financial domain (see [3]).

Similarly, the proposed “two-level learning procedure” is framed as a key innovation, yet such hierarchical or nested optimization schemes—teacher–student, fast–slow, System 1–System 2 (Kahneman), dual-loop, or global–local --- are well-established across AI and reinforcement learning. The use of this structure here appears largely derivative rather than conceptually new.

Consequently, the framework's contribution appears to be more of an effective (and interesting) application and combination of existing methods to a new dataset, rather than the development of a fundamentally novel framework as is claimed.

**Ref**:

1. Kumar, A., Li, Z., Zeng, J., Pathak, D., Sreenath, K., and
Malik, J. Adapting rapid motor adaptation for bipedal
robots. In 2022 IEEE/RSJ International Conference on
Intelligent Robots and Systems (IROS), pp. 1161–1168.
IEEE, 2022.

2. Lee, J., Hwangbo, J., Wellhausen, L., Koltun, V., and Hutter,
M. Learning quadrupedal locomotion over challenging
terrain. Science robotics, 5(47):eabc5986, 2020.

3. Saqur, R. (2024). What Teaches Robots to Walk, Teaches Them to Trade too--Regime Adaptive Execution using Informed Data and LLMs. arXiv preprint arXiv:2406.15508.

**Questions:**

1. Clarification of Novelty:
Many of the ideas presented (in-context adaptation, dual-loop or hierarchical training) closely resemble prior frameworks in meta-learning and robotics adaptation (e.g., [1, 2, 3]). Could the authors clearly articulate what new conceptual element distinguishes AlphaCon beyond its application to financial alpha generation?

2. Relation to Prior Work:
How does AlphaCon substantively differ from other in-context adaptation or dual-level learning frameworks already used in reinforcement learning or financial modeling (e.g., [Saqur 2024](https://arxiv.org/pdf/2406.15508))? A comparison in methodology or results would be valuable.

3. Motivation for Dual-Level Design:
What is the practical or theoretical justification for separating the “Proposer” and “Refiner” stages? Could a single agent with recurrent context (or memory-based meta-adaptation) achieve similar results without added complexity? What about using simple **Retrieval Augmented Generation (RAG )** techniques?

---

### Note · Authors · 2025-11-19

I have read and agree with the venue's withdrawal policy on behalf of myself and my co-authors.